# Occupational Mortality Matrix: A Tool for Epidemiological Assessment of Work-Related Risk Based on Current Data Sources

**DOI:** 10.3390/ijerph19095652

**Published:** 2022-05-06

**Authors:** Stefania Massari, Vittoria Carolina Malpassuti, Alessandra Binazzi, Lorena Paris, Claudio Gariazzo, Alessandro Marinaccio

**Affiliations:** 1Department of Occupational and Environmental Medicine, Epidemiology and Hygiene, Italian National Institute for Insurance against Accidents at Work (INAIL), 00143 Rome, Italy; a.binazzi@inail.it (A.B.); l.paris@inail.it (L.P.); c.gariazzo@inail.it (C.G.); a.marinaccio@inail.it (A.M.); 2Department of Statistical Sciences, University of Rome La Sapienza, 00161 Rome, Italy; vittoriacarolina.malpassuti@uniroma1.it

**Keywords:** causes of death, occupational risk factor, routinely collected health data, proportional mortality ratios

## Abstract

Mortality from occupational diseases significantly afflicts society, in terms of both economic costs and human suffering. The International Labour Organization (ILO) estimated that 2.4 million workers die from work-related diseases every year. In Europe, around 80,000 workers die from cancer attributed to occupational exposure to carcinogens. This study developed the Occupational Mortality Matrix (OMM) aimed to identify significant associations between causes of death and occupational sectors through an individual record linkage between mortality data and the administrative archive of occupational histories. The study population consisted of 6,433,492 deceased subjects in Italy (in the period 2005–2015), of which 2,723,152 records of work histories were retrieved (42%). The proportional mortality ratio (PMR) was estimated to investigate the excess of mortality for specific causes associated with occupational sectors. Higher PMRs were reported for traditionally risky occupations such as shipbuilding for mesothelioma cases (PMR: 8.15; 95% CI: 7.28–9.13) and leather production for sino-nasal cancer (PMR: 5.04; 95% CI: 3.54–7.19), as well as for unexpected risks such as male breast cancer in the pharmaceutical industry (PMR: 2.56; 95% CI: 1.33–4.93) and brain cancer in railways (PMR: 1.43; 95% CI: 1.24–1.66). The OMM proved to be a valid tool for research studies to generate hypotheses about the occupational etiology of diseases, and to monitor and support priority actions for risk reduction in workplaces.

## 1. Introduction

Mortality from occupational diseases and injuries significantly afflicts society, in terms of both economic costs and human suffering. The International Labour Office (ILO) estimates that work-related diseases and accidents account for economic losses as high as 4% of the worldwide gross domestic product [1]. Recent estimates reveal that around 2.4 million workers die from work-related accidents and diseases every year. Circulatory diseases (31%), malignant neoplasms (26%) and respiratory diseases (17%) are the three major illnesses, and together they represent around 75% of total work-related mortality [2]. Occupational cancer has increased considerably in recent decades, mainly due to aging and changes in the proportion of workers exposed. The Global Burden of Disease study estimated 349,000 cancer deaths in 2016 attributable to occupational carcinogens (3.9% of all cancer deaths; 79% in males) worldwide [3]. In Europe, the number of work-related cancers has been estimated to be about 120,000 cases and around 80,000 deaths (around 53% of all work-related deaths) per year [4]. A recent WHO/ILO project presented a Special Issue of a series of 15 systematic reviews, developed to support the estimation of global exposure to occupational risk factors and the attributable burden of disease [5].

Having a clear picture of the impact of work-related deaths is important for monitoring workers’ health, but the diagnosis of occupational diseases is still difficult for many reasons. Firstly, they are clinically indistinguishable from other diseases. In some cases, physicians do not have the knowledge to raise a suspicion of a professional origin, which prevents recognition of the role of occupational exposure [6]. Moreover, the presence of multiple exposures and confounding factors (e.g., lifestyle habits, environmental exposures and individual susceptibility) makes it complex to distinguish the occupational attributable fraction from other risk factors in the explanation of the tumor’s etiology [7,8]. The proportion of occupational diseases in Italy, at the population level, appears to be significantly underestimated. In 2020, the Italian National Institute for Insurance against Accidents at Work (INAIL) reported 15,886 occupational diseases and 912 deceased workers due to a recognized occupational disease [9]. These numbers are far from the estimates available in the scientific literature [10].

There is also the need to define the professional profile of the exposure. The type of exposure to occupational carcinogens has changed over time from a high and prolonged intensity to lower levels and shorter exposure periods. Asbestos exposure has decreased in recent decades due to the ban, while other occupational carcinogens, such as diesel engine exhaust, trichloroethylene, polycyclic aromatic hydrocarbons, chromium, benzene and formaldehyde, are currently widespread in industrial production [11].

For such reasons, it is very valuable to build up information systems able to collect and monitor over time the occurrence of work-related diseases to find new and old risk factors in workplaces.

Many countries have developed epidemiological surveillance systems to estimate work-related illness burdens using various up-to-date data sources (e.g., death certificates, population census data, fiscal archives, contribution and retirement archives, compensation claims for occupational diseases and injury archives) [12].

Through this study, we aimed to test the feasibility in Italy of developing a tool for occupational epidemiological studies based on an individual record linkage between current health data sources (e.g., causes of death) and social security archives. Proportional mortality ratios (PMRs) were calculated for each cause of death and economic sector in order to provide a measure of such associations. The results are presented as the Occupational Mortality Matrix (OMM).

## 2. Materials and Methods

### 2.1. Study Design

The study population was composed of persons aged more than 20 years old who died between 2005 and 2015 and paid social contributions to the National Social Insurance Agency (INPS). Mortality data were linked at the individual level with social security records based on Tax ID codes in accordance with the Italian regulations on privacy. Access to data was granted by the fact that this activity was included in the National Statistics Program (PSN 2017–2019) which provides a prior authorization with the force of law of the Italian Data Protection Authority for statistical surveys of public interest. All epidemiological analyses were performed using anonymous data. The collected data derived from computer files of administrative data acquired from ISTAT (Italian National Statistical Institute) and INPS databases.

Mortality data were collected from administrative sources acquired from ISTAT. Among the available data, causes of death (coded by the ICD-10 classification), age at death and educational level were retrieved. The causes of death were grouped into categories, considering malignant tumors and non-neoplastic diseases. Age at death was classified into ten-year age groups, while educational level was grouped into three ordinal levels (Low, Middle, High).

Occupational data were retrieved from INPS files including individuals who worked in private companies having at least one employee from 1974 onwards. Such a source of data consists of workers employed in private manufacturing, construction, agriculture and the private service sector and includes about 55% of the Italian workforce. Public employment (e.g., ministries, regions, municipalities, armed forces, public schools and universities), self-employed worker, artisan, domestic worker, para-subordinate worker and temporary worker data were not included because such data are not yet integrated in the INPS information system and procedures for the selection and acquisition of social contribution data have not yet been developed. For the whole working history, the available information is the period(s) of employment and the economic activity of the company where the worker was employed. Economic activities were classified according to the Statistical Classification of Economic Activities in the European Community, NACE Rev. 2, grouped into broader categories [13].

The economic sector assigned to each worker was defined according to the duration of employment as long as it was at least 1 year from 1974 onwards. Profession status is not provided by the INPS files.

### 2.2. Statistical Analysis and Occupational Mortality Matrix (OMM)

To examine the pattern of mortality by cause of death and occupational sectors, the proportional mortality ratio (PMR) was calculated. It is obtained by dividing the ratio between deaths due to a specific cause and sector and total deaths in that specific economic sector for the corresponding ratio observed in all economic sectors other than the one considered, as follows:(1)PMRij=DijD.jDij¯D.j¯
where *D_ij_* is the number of deaths for cause *i* and sector *j*;

$D_{.j}\;$ is the number of deaths for all causes in sector *j*;

$D_{i\bar{j}}$ is the number of deaths for cause *i* in all sectors but *j*;

$D_{.\bar{j}}$ is the number of deaths for all causes in all sectors but *j*.

Adjusted PMRs were estimated with a generalized linear model (GLM) using the logarithmic link function and distribution of binary errors to calculate the coefficients of multivariate regression. The 95% confidence intervals (95% Cis) were calculated for each estimate. An offset was added to compute PMRs, and the variables included in the statistical analysis were: cause of death, gender, age, economic sector, period of employment and educational level. Age and educational level were included in the model as control variables.

The period of employment for each economic sector was considered as a proxy of the occupational exposure and categorized into three classes: the ‘longest’ duration of employment greater than 1 year, ‘less than 5 years’ and ‘more than 10 years’. The thresholds were chosen according to the 1st and 2nd (median) percentiles of the distribution of the duration of employment.

The statistical analysis was conducted stratifying by gender and the three periods of employment to better analyze risks in relation to the exposure period.

Adjusted PMRs, organized in a matrix form, were used to perform the Occupational Mortality Matrix (OMM) aimed to estimate associations of causes of death and occupational sectors.

The analyses were performed with the R statistical package.

## 3. Results

The mortality dataset consisted of 6,433,492 deceased subjects (3,123,474 males and 3,310,018 females), and the procedure for matching mortality with occupational data was successful for 2,723,152 workers. The mean total percentage of linkage between mortality and occupational archives amounted to 42% (57% in males, 29% in females). The percentage of linkage was comparable with the proportion of the general population belonging to occupational categories under the INPS Social Security System. Table 1 shows the descriptive statistics of the integrated archive. The average age of the study population was 76 years, with a prevalence of males (65% males versus 35% females) and a low educational level (80%), and most were retired from work (82%). Causes of death were mainly malignant neoplasms (33%) and diseases of the circulatory system (35%), and according to the occupational data, agriculture was the most involved sector (37%), followed by manufacturing industries (30%) and administrative or support service activities (13%).

In Figure 1 and Figure 2, the first 10 economic sectors presenting a statistically significant excess risk of mortality (PMRs > 1, calculated in the economic sector of the longest duration of employment class) are shown by groups of diseases (A) and cancer sites (B) and by gender. A specific picture of the agriculture sector is shown in Figure 3. The number of deaths in males was concentrated in the mechanical and construction industry, mainly for cancers of the lung, digestive, lymphatic and hematopoietic systems and liver. In females, most deaths were observed in administrative activities and in the textile and clothing industry for nervous system diseases and cancer. In the manufacture of clothing apparel, the highest number of fatal cases was observed for lung and colorectal cancer. In Figure 3, the number of deaths with an increased excess of risk (PMR > 1) in agriculture is shown. Cerebrovascular and circulatory system diseases were prevalent for both genders. No cancer sites are displayed because they are associated with a PMR < 1.

The detailed picture of the PMRs is presented in the OMM, considering the three durations of employment (the longest duration, less than 5 years, more than 10 years). The results as a whole are presented in Appendix A; in Table 2 and Table 3 of the manuscript, a selection of PMRs is presented for the sake of brevity.

The highest risks in males typically concerned asbestos-related diseases such as mesothelioma and asbestosis in shipbuilding, railways, the manufacture of refined petroleum products and water transport. This was followed by silicosis in mining and quarrying, the manufacture of ceramic products, the manufacture of concrete articles, cement and plaster, the glass industry and electricity, gas, steam and air conditioning supply, and sino-nasal cancer in the manufacture of wood and leather products. Risk excesses for mesothelioma were found, although at a lower proportion, in females employed in the manufacture of textiles, rubber products, concrete articles, cement and plaster. Significant PMRs were observed for liver cancer in the manufacture of clothing and electricity, gas, steam and air conditioning supply; for male breast cancer, despite the limited cases, in the pharmaceutical and plastic industries; and for brain cancer in railways. Among the diseases, significant risk excesses of mortality were found for respiratory system diseases including asbestosis, silicosis and chronic obstructive pulmonary disease (COPD) in various sectors for males. For females, major risks were observed for COPD in the manufacture of machinery, equipment and motor vehicles and plastics, and for diseases of the circulatory system in agriculture.

The stratified analysis for different durations of employment in each industrial sector showed a visible increase in risks with the duration of occupation for almost all causes of death, but especially for typical occupational diseases (e.g., mesothelioma, sino-nasal, lung, bladder). The results from the 10-year lagged duration analysis confirm the associations evidenced in the OMM for the longest period of employment.

## 4. Discussion

An occupational mortality study, conducted on a large temporal series of observed data and referring to the whole national territory, is not easy to realize due to the difficulty in acquiring information on occupational activity which is often limited or missing, especially with regard to past exposure. To overcome this problem, we used data acquired from current data sources, in particular from ISTAT cause of death records and INPS social security contributions, to collect occupational information such as employing company, industrial sector and duration of employment [14,15] for the entire work history of employees enrolled in the INPS files.

The Occupational Mortality Matrix (OMM), based on integrated data sources, provides a cross-tabulation of fatality risks associated with the economic sector, estimated by PMRs. It may represent a tool for occupational epidemiological studies and provide ‘a priori’ information on the estimated occupational exposure according to the entire working history of the deceased person.

The main results of this study indicate excess mortality risks among men for well-recognized occupational causes of death (e.g., high-etiologic-fraction cancers and asbestos-related diseases). The outcomes were interpreted by comparing them to the available literature in order to verify the validity of the study results.

The highest mortality risks refer to asbestos-related diseases such as mesothelioma, in male workers employed in railways, water transport, the manufacture of basic metals and refined petroleum products and shipbuilding [16,17,18]. Many studies with an ecological or analytical design have estimated the entity of asbestos related to cancer in Italy [19,20,21], confirming risks in the same sectors as presented in this study.

Excess mortality risks were also observed for asbestosis among workers in the manufacture of products such as concrete articles, cement and plaster, in the construction sector and in the building of ships and boats. Heavy exposures historically caused by severe fibrosis with a relatively short latency occurred among the asbestos miners, millers and textile workers. Recently, a study analyzed the asbestosis cases produced by relatively lower exposures but longer latencies (in the period 1960–1970). It found cases in the construction sector (carpenters and joiners, laggers, painters and shop fitters), among pipe fitters, laborers, steel workers and welders in other economic sectors, in naval engineering, in train carriage manufacturing and in power stations, while shipbuilding was not found to be similar to our findings [18].

An excess mortality risk for sino-nasal cancer (SNC) was observed in the manufacture of wood and leather products [22,23]. The incidence of SNC is generally very low but much higher for specific occupational settings and agents [10,24]. In an Italian study, a significant number of SNC cases exposed to leather dust were found, primarily among workers employed in the manufacture or repair of shoes and leather products, tanning processes and retail trade [23].

A high risk of silicosis was described in activities of mining, tunneling and quarrying, the production of ceramics, the construction sector and electricity, gas, steam and air conditioning supply [25,26]. In Italy, many economic activities implicate potential exposures to silica dust associated with mining, digging and rock processing, the production of ceramics and the construction sector (in particular with the renovations industry). Temporal trends and spatial patterns of mortality from silicosis have been recently investigated, and a progressive decline was observed in death rates over time, except for some Italian regions where local occupational activities are still at risk of breathable silica dust exposure [27].

Increased and statistically significant PMRs were found for lung cancer mortality in transport, the manufacture of basic metals, construction, petroleum refinery, fishing and shipbuilding. Such results are consistent with the scientific literature reporting lung cancer mortality in construction due mainly to silica, diesel fumes and asbestos, followed by transportation due to diesel exposures, manufacturing of machinery due to a variety of carcinogen exposures and the basic metal industry [28]. Among petroleum refinery workers, the association of lung cancer is well known considering the exposure to a wide variety of carcinogenic agents such as mixtures of polycyclic aromatic compounds resulting from the incomplete combustion of organic materials, volatile organic compounds or metal compounds of chromium and nickel [29].

Significant associations with colorectal cancer were observed in the petroleum industry due to the wide use of chemical compounds as reported in the literature, and in the sectors of jewelry, railways and repair and installation of machinery, such associations appeared in line with the hypothesis of exposure to asbestos [30,31,32].

The increased risk for bladder cancer in fishing and water transport may be attributed, as known in the literature, to diesel engine emissions [33,34].

A new finding of this study is the increased and statistically significant PMR for male breast cancer found in the pharmaceutical and plastic industries. Even if the disease is rather rare, this finding was confirmed by a European case–control study, which assessed an increased risk of breast cancer in men professionally exposed to chlorinated solvents such as trichloroethylene, usually utilized in the plastic or rubber industry, as well as oxygenated solvents widely used in the paint, pharmaceutical, fragrance, adhesive and food industries [35].

A new association was also observed for brain cancer in railways. Work on trains entails extremely high exposure to low-frequency magnetic fields (EMFs), but this hypothesis, associated with brain cancer, is not supported by the scientific literature [36,37].

A significant increased mortality for circulatory system diseases was observed in this study in agriculture and in the manufacture of tobacco products. The first finding is in line with the literature which shows stress, fatigue and long working hours as probable etiological factors [38]. Instead, in the manufacturing process of tobacco, in the past, it was observed that certain substances such as nicotine, ammonia and carbonic acid were emanated from the fermentation process, causing chronic and acute disorders primarily for diseases of the respiratory and nervous systems and cardiovascular diseases [39].

The strength of this study is that it was possible to place emphasis on female workers who are often omitted in occupational epidemiologic studies due to the low representativeness of this category in the workforce in particular sectors and professions. At present, however, women play a major role in many economic sectors, and gender differences in occupational risk factors are clearly evident in sectors where their presence is relevant (i.e., the textile and shoe industries, health care, personal services and schools). Such differences are noticeable in the manufacture of textiles and clothing and office administrative and other support activities where the number of cancer deaths is higher in women than in men. In textile and clothing production, strong associations were observed for mesothelioma, and bladder, colorectal and kidney cancer. Moreover, for mesothelioma and bladder cancer, the association became stronger for longer periods of employment in such sectors.

The wide body of scientific literature reports well-documented evidence of mortality risks in females for mesothelioma cases in the non-asbestos textile sector, in the chemical and plastic industries [40,41] and among packers, sewing workers, cleaners, canning workers and sorting clerks and postal workers [42]. Risks in the rubber industry, the manufacture of concrete articles, cement and plaster and the jewelry industry are all represented in female occupationally exposed cases of the Italian National Mesothelioma Registry [16].

Higher PMRs were observed for bladder cancer in the textile and plastic industries and in the manufacture of electrical equipment, and mortality risks increased significantly with the duration of employment in the sector. In the literature, no overall excess risk was found in women employed in the textile industry. An excess risk was observed in dye workers, tailors and tailors’ pressers or cutters associated with long-term employment [43], and in workers employed in the plastic industry for chronic infection of the lower urinary tract [44].

Excess risks of lung cancer mortality in women were found in printing and the manufacture of refined petroleum products and basic chemicals, where an association with different widespread carcinogenic agents, from PAHs to metals, is well documented [30].

Further associations in line with the literature referred to increased risks of death from lung and kidney cancer in the manufacture of electrical equipment and for lung and colorectal cancer in the motor vehicle manufacturing industry [45].

The highest mortality among women is due to diseases of the circulatory system mainly associated with stress and long working hours [38]. In this study, the most represented sector, in terms of female fatalities (Figure 1), was administrative, financial and other support activities with an elevated mortality risk for diseases of the nervous system and lung cancer. A recent study carried out in Switzerland confirmed an excess of lung cancer mortality, in both genders among hotel and restaurant workers [46].

### Study Strength and Limitation

The main limitation of this study is that administrative data are not created for scientific or research purposes, and INPS records only report the industrial sector, while exposure intensity and profession are not available. Administrative data cannot be used to characterize the profiles of occupational exposure in an adequate way. The collected information, due to the nature of the data, may not consider confounding factors such as smoking, diet and socioeconomic status. The mortality risk excesses reported here are estimates, so it would be advisable to look further into the specificity of the associations, but this would entail undertaking a detailed but extremely costly investigation. On the contrary, the use of administrative data may fill the gap of the lack of information about the working history of subjects included in epidemiological studies. The present study confirmed most of the known associations between occupation and causes of death and highlighted associations suitable for an in-depth search for past occupational exposure. The integrated system, presented here as the Occupational Mortality Matrix, based on computerized databases, appears to be a promising low-cost method to define ‘a priori’ occupational exposures. Since this procedure is based on mortality data, diseases with low lethality are not highlighted, and the resulting occupational exposure profile reproduces only a situation at risk that occurred in the past. Certainly, hospitalization data could provide a different picture about the current exposure patterns. Moreover, since the matrix is easily achievable, a constant updating of data might be seen as the starting point of an epidemiological surveillance system aimed to detect and monitor hazards in workplaces.

Available data derived from administrative sources provide a well-represented study population over time and space (ten years and national coverage). Occupational data allow tracing the entire work history and not only punctual information related to a specific point in time, e.g., the time of death. Thanks to the high sample size of this study, it was possible to investigate the excess of deaths by cause and industrial sectors for males and females separately.

## 5. Conclusions

The assessment of the burden of fatal diseases in working environments is an important guide for policy makers to identify major risk factors and high-risk populations, and to support decisions on priority actions for risk reduction in the workplace.

A study design based on record linkage procedures among administrative data represents a great opportunity for occupational research studies and can help set priorities for the prevention of occupational diseases. Even for epidemiological studies with an analytical design (e.g., cohort studies focused on environmental risk factors), the use of administrative occupational data is a promising field of action.

## Figures and Tables

**Figure 1 ijerph-19-05652-f001:**
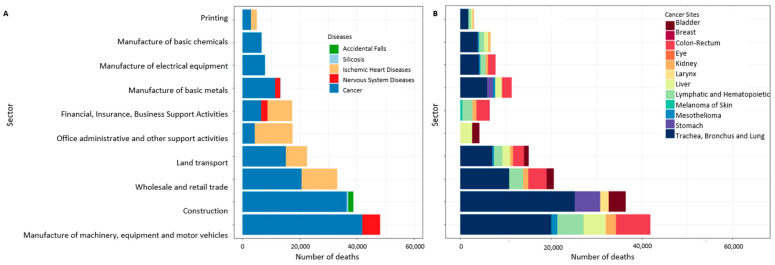
Number of deaths with PMR > 1 by economic sector and diseases or cancer sites in males ((**A**) diseases; (**B**) cancer sites). The PMR was calculated considering the longest duration of employment.

**Figure 2 ijerph-19-05652-f002:**
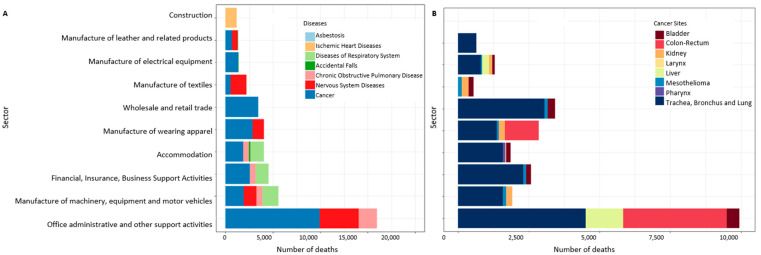
Number of deaths with PMR > 1 by economic sector and diseases or cancer sites in females ((**A**) diseases; (**B**) cancer sites). The PMR was calculated considering the longest duration of employment.

**Figure 3 ijerph-19-05652-f003:**
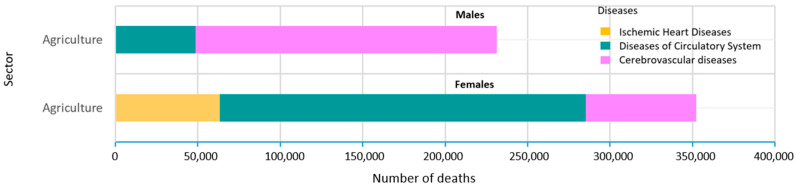
Number of deaths with PMR > 1 in agriculture by diseases and gender (males: 231,155; females: 352,292). The PMR was calculated considering the longest duration of employment.

**Table 1 ijerph-19-05652-t001:** Descriptive data of integrated archive resulting from the record linkage between mortality and occupational data by gender (N = 2,723,152; period: 2005–2015).

Variable	Males		Females	
	N	%	N	%
**Age group (years)**				
20–29	8548	0.48	1866	0.20
30–39	21,280	1.20	8035	0.85
40–49	56,045	3.15	26,498	2.80
50–59	126,493	7.12	57,690	6.10
60–69	271,814	15.29	102,669	10.85
70–79	524,373	29.51	219,384	23.19
≥80	768,678	43.25	529,779	56.01
**Educational level**				
Low (primary school or less)	1,368,305	76.99	798,297	84.39
Middle (secondary school)	281,234	15.82	107,480	11.36
High (high school or university)	127,692	7.18	40,144	4.24
**Causes of death (ICD-10 code)**				
Neoplasms (C00-D49)	649,623	36.55	297,306	31.43
Mental, behavioral and neurodevelopmental disorders (F01-F99)	29,051	1.63	26,164	2.77
Diseases of the nervous system (G00-G99)	57,328	3.23	40,394	4.27
Diseases of the circulatory system (I00-I99)	585,922	32.97	358,320	37.88
Diseases of the respiratory system (J00-J99)	133,710	7.52	49,828	5.27
Injury, poisoning, other consequences of external causes (S00-T88)	84,299	4.74	30,188	3.19
Other causes of death (A00-B99, D50-E89, H00-H59, H60-H95, K00-K95, L00-L99, M00-R99, U00-Z99)	237,298	13.35	143,721	15.19
**Industrial sectors (NACE Rev. 2)**				
Agriculture, forestry and fishing	492,094	27.69	517,068	54.66
Mining and quarrying	11,695	0.66	576	0.06
Manufacturing	612,232	34.45	193,191	20.42
Electricity, gas, steam and air conditioning supply	22,250	1.25	2150	0.23
Construction	232,872	13.10	9646	1.02
Wholesale and retail trade	101,671	5.72	43,796	4.63
Transportation and storage	76,024	4.28	6542	0.69
Human health activities	5852	0.33	12,985	1.37
Waste collection, treatment and disposal activities	4546	0.26	518	0.05
Washing and dry cleaning of textiles	3909	0.22	3478	0.37
Hairdressing, salons	1317	0.07	1635	0.17
Administrative and support service activities	212,769	11.97	154,336	16.32

**Table 2 ijerph-19-05652-t002:** Occupational Mortality Matrix (OMM) based on proportional mortality ratios (PMRs) by cause of death and industrial sector for males.

Cause of Death	Industrial Sectors	Males					
		N	PMR (95% CI)	N_5_	PMR_5_ (95% CI)	N_10_	PMR_10_ (95% CI)
Mesothelioma	Manufacture of refined petroleum products	62	2.38 (1.86–3.05)	8	1.21 (0.60–2.41)	54	2.62 (2.01–3.42)
	Manufacture of basic chemicals	271	1.78 (1.58–2.01)	86	1.75 (1.42–2.16)	185	1.78 (1.54–2.05)
	Manufacture of rubber products	89	1.69 (1.37–2.08)	19	1.29 (0.82–2.02)	70	1.78 (1.41–2.25)
	Manufacture of plastic products	71	1.29 (1.03–1.63)				
	Manufacture of concrete articles, cement and plaster	182	1.42 (1.23–1.65)	55	1.28 (0.98–1.67)	127	1.51 (1.27–1.80)
	Manufacture of basic metals	449	2.05 (1.87–2.25)	107	1.52 (1.26–1.84)	342	2.27 (2.04–2.52)
	Manufacture of machinery, equipment and motor vehicles	1301	1.70 (1.6–1.79)	373	1.54 (1.39–1.71)	928	1.77 (1.65–1.89)
	Manufacture of electrical equipment	259	1.52 (1.35–1.72)	83	1.63 (1.29–1.98)	176	1.47 (1.27–1.70)
	Building of ships and boats	289	8.15 (7.28–9.13)	70	6.51 (5.16–8.21)	219	8.62 (7.57–9.82)
	Manufacture of other products n.e.c.	122	1.68 (1.41–2.01)	64	2.05 (1.61–2.62)	58	1.60 (1.24–2.07)
	Electricity, gas, steam and air conditioning supply	149	1.73 (1.47–2.03)	17	1.73 (1.08–2.78)	132	1.52 (1.28–1.80)
	Railways	131	3.04 (2.53–3.56)	16	2.65 (1.63–4.31)	115	2.73 (2.27–3.28)
	Land transport	365	1.41 (1.27–1.56)	58	0.82 (0.63–1.06)	307	1.58 (1.41–1.77)
	Water transport	120	3.27 (2.73–3.9)	53	3.17 (2.42–4.14)	67	3.92 (3.09–4.97)
Trachea, bronchus and lung	Fishing	955	1.22 (1.15–1.29)	435	1.18 (1.08–1.29)	520	1.28 (1.18–1.39)
	Water transport	1000	1.20 (1.14–1.28)	551	1.22 (1.13–1.32)	449	1.27 (1.16–1.38)
Sino-nasal	Manufacture of wood products	53	4.04 (3.08–5.31)	24	4.42 (2.94–6.63)	29	3.83 (2.65–5.55)
	Manufacture of leather and related products	31	5.04 (3.54–7.19)	15	6.24 (3.74–10.39)	16	4.29 (2.61–7.02)
Larynx	Fishing	68	1.41 (1.11–1.78)	27	1.07 (0.73–1.56)	41	1.70 (1.25–2.31)
	Construction	1845	1.34 (1.28–1.41)	999	1.32 (1.24–1.40)	846	1.38 (1.29–1.48)
Colon-rectum	Photographic activities	50	1.50 (1.15–1.97)	30	1.68 (1.19–2.38)	20	1.45 (0.95–2.21)
	Repairing n.e.c.	41	1.50 (1.11–2.02)	19	1.27 (0.82–1.97)	22	1.91 (1.28–2.86)
	Manufacture of jewellery	108	1.22 (1.02–1.47)	41	1.19 (0.88–1.60)	67	1.25 (0.99–1.58)
	Railways	490	1.31 (1.2–1.42)	69	1.09 (0.87–1.38)	421	1.23 (1.12–1.35)
Liver	Manufacture of clothing	310	1.20 (1.07–1.34)	160	1.27 (1.09–1.48)	150	1.16 (0.99–1.36)
	Electricity, gas, steam and air conditioning supply	587	1.20 (1.11–1.3)	65	0.97 (0.77–1.24)	522	1.19 (1.10–1.30)
Kidney	Fishing	101	1.28 (1.05–1.55)	37	1.03 (0.75–1.42)	64	1.53 (1.20–1.95)
	Printing	221	1.35 (1.19–1.54)	66	1.21 (0.95–1.54)	155	1.38 (1.18–1.62)
	Manufacture of plastic products	157	1.22 (1.05–1.43)	50	1.07 (0.81–1.41)	107	1.29 (1.07–1.56)
Bladder	Fishing	155	1.27 (1.08–1.48)	89	1.33 (1.08–1.63)	66	1.21 (0.95–1.54)
	Water transport	164	1.23 (1.05–1.43)	95	1.13 (0.93–1.39)	69	1.41 (1.12–1.78)
Brain	Railways	171	1.43 (1.24–1.66)	16	1.10 (0.68–1.79)	155	1.22 (1.05–1.43)
Asbestosis	Manufacture of concrete articles, cement and plaster	51	8.11 (6.13–10.71)	25	6.44 (4.31–9.64)	26	10.45 (7.10–15.38)
	Building of ships and boats	48	27.07 (20.34–36.04)	25	23.91 (15.93–35.88)	23	31.64 (21.04–47.59)
Silicosis	Mining and quarrying	240	13.42 (11.76–15.32)	175	11.11 (9.48–13.02)	65	16.91 (13.22–21.61)
	Manufacture of glass and glass products	53	4.08 (3.11–5.34)	33	3.40 (2.41–4.80)	20	5.73 (3.69–8.89)
	Manufacture of ceramic products	102	5.62 (4.61–6.86)	61	4.33 (3.34–5.60)	41	9.00 (6.61–12.27)
	Manufacture of basic metals	162	1.95 (1.66–2.28)	116	1.77 (1.46–2.14)	46	2.29 (1.65–2.94)
	Electricity, gas, steam and air conditioning supply	100	2.93 (2.40–3.58)	38	3.81 (2.75–5.26)	62	4.13 (3.19–5.35)
	Construction	485	1.49 (1.35–1.64)	373	1.28 (1.14–1.43)	112	1.53 (1.26–1.85)
Diseases of circulatory system	Manufacture of tobacco products	145	1.23 (1.09–1.38)	70	1.10 (0.92–1.31)	75	1.40 (1.19–1.64)
Cerebrovascular diseases	Manufacture of tobacco products	43	1.40 (1.06–1.84)	22	1.31 (0.89–1.92)	21	1.52 (1.02–2.26)
Diseases of respiratory system	Mining and quarrying	1612	1.55 (1.48–1.62)	1119	1.70 (1.61–1.79)	493	1.27 (1.17–1.39)
Chronic obstructive pulmonary disease	Mining and quarrying	762	1.43 (1.34–1.53)	544	1.58 (1.46–1.71)	218	1.14 (1.00–1.29)
Nervous system diseases	Manufacture of pharmaceutical preparations	321	1.25 (1.12–1.39)	104	1.15 (0.95–1.38)	217	1.30 (1.14–1.48)

Note: PMRs are displayed if greater than 1.20 and N. deaths > 30. PMRs are adjusted for age and educational level. Age > 20 years. Abbreviations: N: deceased people employed in the sector with the longest duration of employment; PMR: proportional mortality ratio calculated on the longest period of employment; CI: 95% confidence interval; N_5_: deceased people employed in the sector for less than 5 years; PMR_5_: proportional mortality ratio calculated on the period of employment in the sector for less than 5 years; N_10_: deceased people employed in the sector for more than 10 years; PMR_10_: proportional mortality ratio calculated on the period of employment in the sector for more than 10 years.

**Table 3 ijerph-19-05652-t003:** Occupational Mortality Matrix (OMM) based on proportional mortality ratios (PMRs) by cause of death and industrial sector (*) for females.

Cause of Death	Industrial Sectors	Females					
		N	PMR (95% CI)	N_5_	PMR_5_ (95% CI)	N_10_	PMR_10_ (95% CI)
Mesothelioma	Manufacture of textiles	136	2.20 (1.86–2.60)	74	1.78 (1.42–2.24)	62	2.82 (2.18–3.65)
	Manufacture of clothing apparel	80	1.38 (1.10–1.72)	39	1.11 (0.81–1.52)	41	1.75 (1.27–2.41)
	Manufacture of basic chemicals	24	1.81 (1.22–2.70)	15	1.96 (1.18–3.24)	9	1.55 (0.81–2.98)
	Manufacture of rubber products	15	2.74 (1.65–4.53)	10	3.15 (1.70–5.83)	5	2.08 (0.86–4.99)
	Manufacture of wood products	24	1.60 (1.07–2.39)	11	1.18 (0.65–2.13)	13	2.21 (1.28–3.83)
	Manufacture of basic metals	18	1.98 (1.25–3.13)	13	2.50 (1.45–4.29)	5	1.24 (0.52–2.98)
	Manufacture of machinery, equipment and motor vehicles	109	1.77 (1.46–2.14)	52	1.57 (1.19–2.06)	57	1.96 (2.57–1.50)
	Manufacture of electrical equipment	45	1.50 (1.12–2.01)	20	1.31 (0.84–2.02)	25	1.65 (1.11–2.46)
	Wholesale and retail trade	127	1.25 (1.04–1.50)	63	1.06 (0.82–1.37)	64	1.51 (1.16–1.96)
	Financial, insurance activities and business support service activities	105	1.30 (1.06–1.58)	47	1.05 (0.78–1.41)	58	1.63 (1.23–2.15)
Trachea, bronchus and lung	Printing	353	1.56 (1.41–1.73)	167	1.41 (1.22–1.63)	186	1.70 (1.48–1.95)
	Manufacture of refined petroleum products	48	1.77 (1.35–2.31)	16	1.33 (0.82–2.13)	32	2.07 (1.50–2.87)
	Manufacture of basic chemicals	349	1.24 (1.12–1.37)	155	0.96 (0.82–1.12)	194	1.54 (1.34–1.76)
	Wholesale and retail trade	3052	1.32 (1.27–1.37)	1588	1.16 (1.10–1.22)	1464	1.51 (1.43–1.59)
	Accommodation	1593	1.21 (1.15–1.27)	1136	1.10 (1.04–1.16)	457	1.59 (1.45–1.74)
Colon-rectum	Manufacture of refined petroleum products	38	1.59 (1.17–2.16)	14	1.35 (0.82–2.25)	24	1.67 (1.14–2.45)
Kidney	Manufacture of electrical equipment	113	1.22 (1.01–1.46)	49	1.10 (0.83–1.46)	64	1.31 (1.02–1.67)
	Manufacture of wood products	63	1.36 (1.06–1.74)	35	1.30 (0.93–1.81)	28	1.48 (1.02–2.14)
Bladder	Manufacture of textiles	171	1.25 (1.07–1.45)	112	1.12 (0.93–1.35)	59	1.43 (1.10–1.84)
	Manufacture of plastic products	34	1.50 (1.07–2.10)	15	1.03 (0.62–1.71)	19	2.23 (1.42–3.50)
	Manufacture of electrical equipment	82	1.38 (1.11–1.71)	47	1.43 (1.07–1.90)	35	1.28 (0.92–1.78)
Diseases of circulatory system	Agriculture	224,422	1.17 (1.17–1.18)	73,914	1.04 (1.03–1.04)	150,508	1.43 (1.43–1.44)
Ischemic heart diseases	Agriculture	62,262	1.10 (1.09–1.11)	20,957	1.00 (0.99–1.01)	41,305	1.32 (1.31–1.34)
Cerebrovascular diseases	Agriculture	65,608	1.16 (1.15–1.17)	21,468	1.02 (1.01–1.04)	44,140	1.43 (1.42–1.45)
Diseases of respiratory system	Manufacture of machinery, equipment and motor vehicles	1663	1.09 (1.04–1.14)	1047	1.08 (1.02–1.14)	616	1.09 (1.01–1.17)
	Manufacture of rubber products	197	1.24 (1.09–1.42)	139	1.28 (1.09–1.50)	58	1.13 (0.88–1.46)
Chronic obstructive pulmonary disease	Manufacture of machinery, equipment and motor vehicles	668	1.18 (1.10–1.27)	416	1.12 (1.02–1.23)	252	1.24 (1.10–1.40)
	Manufacture of plastic products	126	1.21 (1.02–1.44)	85	1.12 (0.91–1.39)	41	1.31 (0.96–1.77)
Nervous system diseases	Manufacture of clothing apparel	1188	1.11 (1.05–1.18)	785	1.12 (1.05–1.2)	403	1.08 (0.98–1.19)

Note: PMRs are displayed if greater than 1.20 and N. deaths > 15 for neoplasms and if greater than 1.10 and N. deaths > 30 for diseases. PMRs are adjusted for age and educational level. Age > 20 years. Abbreviations: N: deceased people employed in the sector with the longest duration of employment; PMR: proportional mortality ratio calculated on the longest period of employment; CI: 95% confidence interval; N_5_: deceased people employed in the sector for less than 5 years; PMR_5_: proportional mortality ratio calculated on the period of employment in the sector for less than 5 years; N_10_: deceased people employed in the sector for more than 10 years; PMR_10_: proportional mortality ratio calculated on the period of employment in the sector for more than 10 years.

## Data Availability

Not applicable.

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
