# Peer review of "Occupational Mortality Matrix: A Tool for Epidemiological Assessment of Work-Related Risk Based on Current Data Sources"

_ijerph, 2022, doi:10.3390/ijerph19095652_

Round 1
Reviewer 1 Report
Interesting and useful, at least as a starting point in occupational diseases.
In my opinion, the paper is quite useful because the study and the prove of occupational diseases is a matter that needs much more analysis, particularly, disaggregation by sex and occupational position, and this study gives a general idea. It lacks the disaggregation by occupational position (job classification) but it is useful and its conclusions seem coherent with the current general feeling.
It is interesting, as well, that the authors have worked with administrative data and it shows the important information they can provide.
Also, I can mention the conclusions on vascular diseases related with administrative positions and jobs and the genre perspective in those cases.
Author Response
Reviewer #1:
Interesting and useful, at least as a starting point in occupational diseases.
In my opinion, the paper is quite useful because the study and the prove of occupational diseases is a matter that needs much more analysis, particularly, disaggregation by sex and occupational position, and this study gives a general idea. It lacks the disaggregation by occupational position (job classification) but it is useful and its conclusions seem coherent with the current general feeling.
It is interesting, as well, that the authors have worked with administrative data and it shows the important information they can provide.
Also, I can mention the conclusions on vascular diseases related with administrative positions and jobs and the genre perspective in those cases.
We thank the reviewer for these comments. We agree that the lack of disaggregation by occupational position is extremely useful for analytical studies but this information is not available in INPS files.
Reviewer 2 Report
This is a well planned and executed work aimed at assessing work-related risk of death using occupational mortality matrix based on proportional mortality ratio on publicly available data linked with occupational data in Italy between 2005 and 2015. The authors reported increased excess death due to various cancer types diseases associated with work-related exposure to carcinogens and risk factors respectively. However, there are some factors that needs to be addressed.
MAJOR
- The manuscript needs extensive English language proofing aimed at improving clarity, concision, and quality of discussion. Extra focus should be directed toward the discussion section.
- Lines 49-52: Is the limitation of not being able to distinguish the aetiology of neoplasm about training and experience or equipment? I don't think training or experience could resolve such fundamentally technical/biochemical issue.
I suggest the authors to look more into the topic of multi-factorial etio-pathogenesis and support this discussion with relevant literature especially since the direction of this will define how serious the findings in this study will be taken by readers.
- Line 82: Please provide justification for using data or persons above 20 years old. Is that the legal working age in Italy?
- Line 151-154: Could some of the deaths be related more to lifestyle rather than occupation. For instance, liver cancer is associated with chronic cirrhosis which may be related to alcohol consumption which is disproportionately prevalent in men. More attention should be paid to this in both introduction and discussion section to help readers understand the various cofounders that may influence the findings and interpretation of it.
- Lines 177-178: Manufacture of refined petroleum product and others seems to overlap as a risk factors. This is a limitation with significant implication on how the result is interpreted.
The authors should discuss this in the limitation section.
- Discussion should ideally start with the major findings of the study and then the context of the finding in terms of previous reports
- Line 254: Although not entirely clear what the authors are trying to say here, the use of the term ‘causal link’ should hardly be used as association is not causation and even well-designed studies will struggle to established causation and can only establish correlation etc. The problem of causation is a complex mathematical topic that will require the knowledge of Granger causality, transfer entropy etc. and beyond the scope of this study.
- Conclusion should give overall findings and suggestions etc. not repeat what has been said in the discussion.
MINOR
- Title a bit vague and could benefit from being a bit more specific. For instance, instead of "...a tool for epidemiological studies based on current data sources" it could be replaced by "...a tool for epidemiological assessment of work-related risk ".
- Line 14: "...died from cancer...".
- Line 17: 'amounts to' sounds better as 'represents'.
- Lines 18-19: Reads better as '...Italy (in the period 2005-2015) of which 2,723,152 (42.3%) records of work histories were retrieved. '
- Line 20: 'associated with'
- Lines 20-21: Reads better as '...for traditionally risky occupations such as shipbuilding...' or please clarify.
- Line 25: 'research' not 're-search'.
- Lines 41-43: What measure? cancer death or work-related death? please clarify and proofread the sentence to make sure the point aimed is clear.
- Line 53: ' The proportion of occupational...'
- Lines 56-57: Please provide relevant literatures being referred to here.
- Line 68: current or various?
- Line 81: '...recorded between 2005-2015 for persons...'
- Line 85: Please spell out the full meaning of INPS as its appearing for the first time here.
- Line 88: '...private services sector...'
- Lines 88-90: Please provide brief details of why public employees were not included
- Lines 93-94: Please provide reference
- Line 116: Please spell out GLM (generalised linear model) as not all readers are able to tell what you meant.
- Lines 120-121: Please clarify what you mean by 'title of study'?
- Line 139: shows the
- Cause of death data in table 1 not aligned
- Line 150: shown
- Line 171: The meaning of prevalence here is not clear, please define?
- Line 172: 'shown' not showed.
- Line 181-182: 'Similar risk excesses for mesothelioma was found, although in lower proportion ...'
- Line 184: '...neoplasms of digestive organs and for breast cancer mortality in...'
- Line 191: '...showed a visible positive trend of risks with the duration of occupation for almost all causes of...'
- Lines 194-196: The message here is not clear because of the length of the sentence (too long). The authors should possibly break into two or simplify for clarity.
- Table 2: Colon-rectum sounds better as colorectal
- Line 279: ‘commonly’ I assume you mean ‘similar to findings here’?
- Line 321: sentence is absolutely unclear.
Author Response
Reviewer #2:
This is a well planned and executed work aimed at assessing work-related risk of death using occupational mortality matrix based on proportional mortality ratio on publicly available data linked with occupational data in Italy between 2005 and 2015. The authors reported increased excess death due to various cancer types diseases associated with work-related exposure to carcinogens and risk factors respectively. However, there are some factors that needs to be addressed.
MAJOR
- The manuscript needs extensive English language proofing aimed at improving clarity, concision, and quality of discussion. Extra focus should be directed toward the discussion section.
Following the reviewer’s suggestion we have carried out an extensive and accurate revision of English language asking for the advice of a native speaker.
- Lines 49-52: Is the limitation of not being able to distinguish the aetiology of neoplasm about training and experience or equipment? I don't think training or experience could resolve such fundamentally technical/biochemical issue. I suggest the authors to look more into the topic of multi-factorial etio-pathogenesis and support this discussion with relevant literature especially since the direction of this will define how serious the findings in this study will be taken by readers.
We agree with the reviewer that training and experience are not important to distinguish the aetiology of neoplasm. We meant to say that experience of physicians is essential to raise a suspicion of work-related origin and lead to an in-depth investigation to understand the exact role of the occupational exposure. The sentence was changed in order to make the concept clearer.
- Line 82: Please provide justification for using data or persons above 20 years old. Is that the legal working age in Italy?
In Italy, the minimum age to be admitted to work is 16, but in the class 16-19 years the number of deaths was too small to be modeled (<5%). For this reason the manuscript was not changed.
- Line 151-154: Could some of the deaths be related more to lifestyle rather than occupation. For instance, liver cancer is associated with chronic cirrhosis which may be related to alcohol consumption which is disproportionately prevalent in men. More attention should be paid to this in both introduction and discussion section to help readers understand the various cofounders that may influence the findings and interpretation of it.
We agree that the role of lifestyle is critical in the aetiology of cancer and cannot be managed due to the lack of this information in the dataset. We add a sentence in the introduction and in the paragraph of study limitation.
- Lines 177-178: Manufacture of refined petroleum product and others seems to overlap as a risk factors. This is a limitation with significant implication on how the result is interpreted. The authors should discuss this in the limitation section.
The overlap of some sectors as risk factor is due to a recognized presence of asbestos in both refined petroleum sector as well as shipbuilding and railways. We changed the sentence to better explain the concept.
- Discussion should ideally start with the major findings of the study and then the context of the finding in terms of previous reports
We thank the reviewer for his/her suggestion. The paragraph of the “Discussion” was revised in a more organized way.
- Line 254: Although not entirely clear what the authors are trying to say here, the use of the term ‘causal link’ should hardly be used as association is not causation and even well-designed studies will struggle to established causation and can only establish correlation etc. The problem of causation is a complex mathematical topic that will require the knowledge of Granger causality, transfer entropy etc. and beyond the scope of this study.
We thank the reviewer for his/her valuable comments. We have modified the sentence according to the reviewer’s suggestion.
- Conclusion should give overall findings and suggestions etc. not repeat what has been said in the discussion.
Following the reviewer’s suggestion, we have modified the “Conclusions” section particularly as regards future research perspectives.
MINOR
- Title a bit vague and could benefit from being a bit more specific. For instance, instead of "...a tool for epidemiological studies based on current data sources" it could be replaced by "...a tool for epidemiological assessment of work-related risk ".
According to the reviewer’s suggestion, the title was changed.
- Line 14: "...died from cancer...".
Ok
- Line 17: 'amounts to' sounds better as 'represents'.
The sentence was changed according to the reviewer’s comment.
- Lines 18-19: Reads better as '...Italy (in the period 2005-2015) of which 2,723,152 (42.3%) records of work histories were retrieved. '
We have made the changes suggested by the reviewer
- Line 20: 'associated with'
Ok
- Lines 20-21: Reads better as '...for traditionally risky occupations such as shipbuilding...' or please clarify.
Ok
- Line 25: 'research' not 're-search'.
Ok
- Lines 41-43: What measure? cancer death or work-related death? please clarify and proofread the sentence to make sure the point aimed is clear.
Measure was referred to work related cancer cases. We have made changes according to the reviewer’s comment.
- Line 53: ' The proportion of occupational...'
We have made the changes suggested by the reviewer
- Lines 56-57: Please provide relevant literatures being referred to here.
Reference was added.
- Line 68: current or various?
Following the reviewer’s suggestion the phrase was changed in ‘various up-to-date data sources’
- Line 81: '...recorded between 2005-2015 for persons...'
Ok
- Line 85: Please spell out the full meaning of INPS as its appearing for the first time here.
Ok
- Line 88: '...private services sector...'
We have made the changes suggested by the reviewer
- Lines 88-90: Please provide brief details of why public employees were not included
In Italian Institute for Social Security (INPS) there are different funds according to the categories of workers: employees of the private sector signed up to the Employed Workers Pension Fund (FLPD), employees in the public sector, independent workers and self-employed workers. Procedures for the acquisition of the social contribution data used in this study were established for the fund of employees of the private sector only. Integration with the others funds is under development. A sentence was added to better clarify the concept.
- Lines 93-94: Please provide reference
We have made the changes suggested by the reviewer
- Line 116: Please spell out GLM (generalised linear model) as not all readers are able to tell what you meant.
We have made the changes suggested by the reviewer
- Lines 120-121: Please clarify what you mean by 'title of study'?
Thanks to the reviewer’s suggestion we changed the term 'title of study' in ‘educational level’.
- Line 139: shows the
We have made the changes suggested by the reviewer
- Cause of death data in table 1 not aligned
We have made the changes suggested by the reviewer and now the table 1 is aligned
- Line 150: shown
According to the reviewer’s suggestion the term was uniformed in the whole manuscript
- Line 171: The meaning of prevalence here is not clear, please define?
Prevalence refers to the longest duration of the employment. The term was changed in the manuscript.
- Line 172: 'shown' not showed.
According to the reviewer’s suggestion the term was aligned in the whole manuscript
- Line 181-182: 'Similar risk excesses for mesothelioma was found, although in lower proportion ...'
We have made the changes suggested by the reviewer
- Line 184: '...neoplasms of digestive organs and for breast cancer mortality in...'
We have made the changes suggested by the reviewer
- Line 191: '...showed a visible positive trend of risks with the duration of occupation for almost all causes of...'
We have made the changes suggested by the reviewer
- Lines 194-196: The message here is not clear because of the length of the sentence (too long). The authors should possibly break into two or simplify for clarity.
Sentence was changed as suggested by the reviewer
- Table 2: Colon-rectum sounds better as colorectal
We have made the changes as suggested by the reviewer
- Line 279: ‘commonly’ I assume you mean ‘similar to findings here’?
We have made the changes as suggested by the reviewer
- Line 321: sentence is absolutely unclear.
We have made the changes as suggested by the reviewer

Reviewer 3 Report
Please see the attachment.

Author Response
Reviewer #3:
The objective of this study is the development of the Occupational Mortality Matrix (OMM) to identify significant associations between causes of death and occupational sectors through an individual record link between mortality data and the administrative file of occupational histories. . This is an interesting manuscript and well-developed research, but the following revisions and modifications are proposed:
Resume: The authors must separate the different sections (Introduction, methods, results, conclusions).
Abstract was structured following the instruction for authors stated by the journal.
Introduction: Line 34: "Recent estimates revealed that about 2.4 million workers die each year from work-related illnesses." 2.4 million?
The figure was retrieved from literature as mentioned in the reference. It refers to a world-wide estimate.
Methods: The authors should specify the study population, access to data, the data collection and analysis process, the variables analyzed... and explain it in a more organized way.
The manuscript was revised to better explain the methodological phases.
Results: Decimals always go with "." and not with ",". Authors should always put the same number of decimals.
According to the reviewer’s suggestion, format of decimals was revised in the whole manuscript.
Conclusions: In this section, the main findings of the study should appear in a synthesized form
Following the reviewer’s suggestion, we have modified the “Conclusions” section particularly as regards future research perspectives.

Round 2
Reviewer 2 Report
No further comments